# Update on Adjustable Trans-Obturator Male System (ATOMS) for Male Incontinence after Prostate Cancer Surgery

**Carlos Téllez** [1,2], **Juliusz Szczesniewski** [1,2], **Miguel Virseda-Chamorro** [3], **Ignacio Arance** [1,2] **and Javier C. Angulo** [1,2,*]

1. Clinical Department, Faculty of Biomedical Sciences, Universidad Europea, 28805 Madrid, Spain
2. Urology Department, Hospital Universitario de Getafe, 28805 Madrid, Spain
3. Urology Department, Hospital Nacional de Parapléjicos, Carretera de la Peraleda, S/N, 45004 Toledo, Spain
* Correspondence: javier.angulo@universidadeuropea.es

**Abstract:** (1) Background: The adjustable trans-obturator male system (ATOMS) is a surgical device developed to treat post-prostatectomy incontinence (PPI) after prostate cancer treatment. We review the current literature on this anti-incontinence device with the intention of assessing the effectiveness, safety and duration of the silicone-covered scrotal port (SSP) ATOMS, the only generation of the device that is currently available. (2) Material and Methods: Non-systematic literature review is performed. Forty-eight full-text articles are assessed for eligibility. Case reports, expert opinions or commentaries without specific data reported (*n* = 6), studies with patients who underwent intervention before 2014 (IP or SP ATOMS; *n* = 10), and studies with incontinence after transurethral resection of the prostate (TUR-P; *n* = 2) are excluded for analysis. Thirty studies with SSP ATOMS are included in a qualitative synthesis that incorporates systematic reviews (*n* = 3), articles partially overlapping with other previously published studies (e.g., follow-up or series updates; *n* = 9), and studies focusing on specific populations (*n* = 8). Only articles revealing outcomes of SSP ATOMS were included in the quantitative synthesis of results (*n* = 10). (3) Results: the pooled data of 1515 patients from the 10 studies with SSP ATOMS confirmed very satisfactory results with this device after adjustment: dry rate: 63–82%, improved rate: 85–100%, complication rate: 7–33%, device infection rate: 2.7–6.2% and explant rate: 0–19%. The durability of the device is reassuring, with 89% of devices in place 5 years after implantation. (4) Conclusion: Despite the absence of randomized controlled studies, the literature findings confirm results of SSP ATOMS appear equivalent to those of artificial urinary sphincters (AUSs) in terms of continence, satisfaction and complications, but with a lower rate of revision in the long-term. A prospective study identified that patients with daily pad test results <900 mL and a Male Stress Incontinence Grading Scale (MSIGS) of not 4 (i.e., early and persistent stream or urine loss) are the best candidates. Future studies centered on the elder population at higher risk of impaired cognitive ability and in patients including radiation as prostate cancer treatment are needed.

**Keywords:** adjustable trans-obturator male system; effectiveness; safety; post-prostatectomy incontinence; artificial urinary sphincter; prostate cancer

## 1. Post-Prostatectomy Incontinence (PPI) Treatment Options

Radical prostatectomy is an oncologic curative procedure that implies complete removal of the prostate, which may result in urethral sphincteric damage and, consequently, PPI that negatively affects the patient's quality of life [1]. Older age of the patient, non-sparing neurovascular resection and stricture of the anastomosis are risk factors for PPI [2]. Additionally, the preoperative length of the membranous urethra evaluated by magnetic resonance imaging can predict the risk of developing PPI [3].

The restriction of excess water intake and pelvic floor muscle training are the first-line treatment options, but when significant PPI persists, prosthetic surgery is indicated [4,5].

The artificial urinary sphincter has been the only device for many years, and it is still considered the gold standard, despite the significant risk of complications leading to device explantation [6–8]. More recently, the use of male slings to improve urethral sphincter function by repositioning the posterior urethra offers a less invasive surgical approach, often similarly effective as an artificial urinary sphincter (AUS) and with a lower rate of complications [9,10].

The adjustable trans-obturator male system is a modified sling, increasingly used in Europe and Canada in the last decade [11]. This device consists of a tape-shaped mesh implant with a central integrated silicone cushion placed under the bulbar urethra and connected to an access port for perioperative filling that is placed in the scrotum for optimal postoperative adjustment of the cushion volume. The adjustable trans-obturator male system (ATOMS) exerts as a sling that reinforces the urethral sphincter by compression of the bulbar urethra without the need for manipulation by the patient. However, to exert its role ATOMS requires some residual sphincteric function [12].

The best ATOMS results can be expected in mild-moderate PPI and in patients without radiation [13]. However, several other circumstances favor the implant of ATOMS over AUSs, such as impaired cognitive status, limited manual dexterity and unwillingness to repeat a previously failed AUS [5,14]. Possibly the main advantage of ATOMS is easily accessible postoperative adjustment through percutaneous puncture of the scrotal port. Unlike non-adjustable retrobulbar slings, the ATOMS can be an option even in radiated patients and also in selected patients with moderate and moderate-to-severe PPI, provided some residual sphincteric function exists [10]. ATOMS is an effective and safe second-line treatment option for recurrent urinary incontinence after implantation of an AdVance/AdVance XP fixed male sling [15].

However, ATOMS is not the only incontinence device that can be adjusted postoperatively. Other adjustable systems, such as adjustable continence therapy (ProACT) system, male readjustment mechanical external (REEMEX) system, Argus sling and several types of AUS, are available options. All of them intend to improve the results of simple retrobulbar slings and, at the same time, avoid urethral atrophy and erosion that can be expected with the long-term use of the AUS [1]. Their mode of action differs, and their effectiveness and the risk of complications [16]. In the absence of randomized comparative evaluations available, systematic review and meta-analysis allow indirect comparison [17,18]. Generally speaking, adjustable devices have increased the extent of PPI severity to be treated with the trans-obturator technique. Also, differences among adjustable devices allow a more personalized approach to PPI.

## 2. Development and Evolution of the ATOMS Device

The ATOMS® device was developed by Agency for Medical Innovations GmbH (Feldkirch, Austria). This device allows the bulbar urethra to be compressed only on one side and not circumferentially like the AUS. Ventral urethral compression can be increased after surgery by percutaneous injection of sterile saline solution into the port without intervention [11,12]. A prospective comparative study between the ATOMS and the AUS confirmed equivalent efficacy and patient satisfaction results, with a lower revision rate and higher durability for ATOMS [10]. It should be noted, however, the experience worldwide with AUSs is much more extensive.

The ATOMS concept was initiated as a self-anchoring trans-obturator male sling with a cadaveric study published in 2005 [19]. The device developed in 2008 used a titanium circular port placed in the inguinal region and required a double incision, perineal and inguinal, to be placed [20,21]. A modification of the system in 2013 took to place the port in the scrotum and cover it with a membrane, thus avoiding the inguinal incision but still requiring a connection. In 2014 the last modification changed to a pre-attached scrotal port with a silicone cover [22]. The trans-obturator passage of the mesh and the perineal location of the cushion under the bulbar urethra are the same in the three generations of devices. However, the evolution of the inguinal-to-scrotal port has allowed one to perform

the implant with a single incision, thus lowering the risk of infection, diminishing pain and facilitating postoperative adjustment [11]. Also, the pre-attached design avoids the need for the tube connection needed, shortens the operative time of the procedure and avoids the risk of serum leakage. Also, the silicone-covered scrotal port (SSP) avoids the titanium reaction, which, although rare, was observed in some patients with the first and second generation of the device. Since 2014 the ATOMS has remained unchanged, and the only recent modification is the availability of two differently designed helical needles to pass the obturator foramen adapting better to the anthropometric differences like the size of the ischial bone and body mass index.

The interest in the use of ATOMS has increased within the last decade in Europe, Canada, Asia and South and Central America as the experience with the device has broadened the classical indication of male retro-urethral slings from mild-to-moderate to moderate-to-severe incontinence; and also, in selected cases, to patients with severe incontinence [12].

## 3. Mode of Action of ATOMS

ATOMS increases residual sphincteric function to achieve continence by increasing intraurethral pressure, secondary to the stretching effect on the urethral wall caused by cushion filling, which increases urethral resistance [23]. The tension is increased proportionally to the urethral stretching effect produced by the fibrotic capsule in the periphery of the ATOMS cushion. Elastin fibers and muscle are tissues with high compliance and allow great deformations without a proportional boost of tension. Collagen fibers allow initial deformation without high tension, but once the working length is reached, they resist further stretching by sharply increasing their tension. This is the response of a diseased tissue that has been radiated or that has suffered urethral or anastomotic stricture.

Intraoperative urethral pressure measurement at different fillings of the implanted ATOMS cushion confirms two different patterns. Normal urethras with predominant elastin and muscle fibers behave differently than rigid urethras rich in collagen. In more fibrotic urethrae overfilling of the ATOMS cushion is not proportional to continence achieved. In fact, it has already been observed that the postoperative filling of ATOMS tends to be higher in patients who do not reach continence [24]. That also explains why patients with radiation are less likely to achieve full continence [24–26]. The urethral stretching effect is more physiologic than other possible mechanisms which act by kinking and obstructing the urethra. The response to serial filling performed intraoperatively depends on urethral rigidity, reflects baseline SUI severity, and predicts postoperative results [23].

Prospective evaluation of a case series with urodynamic studies performed before and after ATOMS implantation confirmed that direct ventral compression of the bulbar urethra by the silicone cushion significantly increased urethral resistance, which is abnormally low in males with PPI. The bladder outlet obstruction index (BOOI) did not reach a pathological level so as to reflect *de novo* obstruction [27]. The anteroposterior urethral diameter decreased, and the membranous urethra lengthened with a serial increment of the volume of ATOMS cushion, with narrowing of the urethral lumen in the distal part of the rhabdo-sphincter (Figure 1). Therefore, a double mode of action of ATOMS can be expected to contribute to sphincteric reinforcement. That includes the direct compression of the penile and bulbar urethra and elongation of the membranous urethra.

No obstructive effect or relevant voiding phase dysfunction has been detected in another longitudinal prospective study performed in patients implanted with an ATOMS for more than a year, comparing baseline and follow-up urodynamics. Urethral resistance factor (URA) was under 29 cm $H_2O$ in all cases postoperatively, thus, confirming ATOMS does not cause obstruction [28]. This observation is consonant with long-term clinical data with ATOMS in which chronic voiding dysfunction was absent at a 5-year mean follow-up [29]. Preoperative functional voiding parameters are also useful to anticipate the postoperative outcome of ATOMS implants. Higher bladder capacity, lower maximal voiding abdominal pressure and lower BOOI baseline favor continence results postoperatively [30].

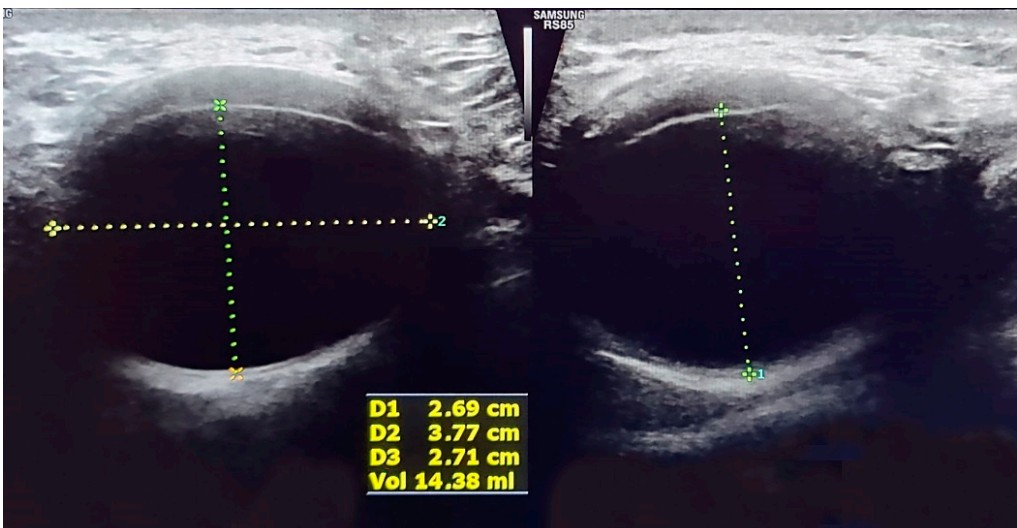

**Figure 1.** Lineal perineal ultrasound shows cushion volume filling and ventral compression with narrowing of the urethral lumen and length increase in the longitudinal section (right side image).

## 4. Surgical Technique of ATOMS

The ATOMS system is an adjustable trans-obturator male sling that consists of a tape-shaped mesh implant with a central integrated cushion and an access port in the scrotum for the adjustment of the cushion volume. Implantation is performed in patients with negative urine cultures and under dual antibiotic treatment, consisting of one single-shot i.v. administration of 240 mg gentamicin 2 h before induction and three times a day oral administration of 500/125 mg amoxicillin/clavulanic acid for 1 week postoperatively.

Under spinal anesthesia, the patient is placed in the lithotomy position. A 14 French catheter is inserted before the procedure. Perineal median incision opens the Colles fascia to expose the *bulbospongiosus* muscle without dissection. The fossa ischiorectal is exposed on both sides with the identification of ischiocavernosus muscles. After that, trans-obturator needles are used to pass the mesh from outside in. The passing needle, available in two shapes that adapt to the anatomical variation of the ischial bone (Figure 2a), is moved around the ramus inferior of the pubic bone by means of a rotary movement using the index finger to identify the tip of the needle in the fossa ischiorectal and guide the suture that brings the mesh out of the surgical wound. Then, the needle is pulled back, and the implant is brought into position by pulling the sling arms. With this maneuver, the cushion gently compresses the urethra ventrally (Figure 2b). After the sheath covers of the sling arms are removed, the mesh arms are secured to the central cushion under tension on both sides using the integrated non-absorbable attachment sutures, thus establishing a four-point fixation in the male pelvis that avoids dislocation of the cushion (Figure 2c).

Intraoperative washing with an antibiotic solution is recommended. The cushion is vented by incorporating and extracting saline solution. Again, the physiological sodium chloride solution is incorporated up to a regular atmosphere pressure, which usually corresponds to 6–8 mL, depending on how firm the suture was knotted. Intraoperative overfilling of the cushion can be incorporated in correlation with the baseline severity of the 24-h pad test. The optimal positioning of the cushion is important (Figure 2d). Central placement is required to avoid dislocation that can be caused by a non-symmetrical obturator passage or by unequal tensioning of the sutures on each side. When the cushion is not symmetrically placed, ATOMS filling will cause lateral and not ventral compression of the urethra, which can negatively affect its efficacy. Discrete bleeding after mesh passage is sometimes observed, but oozing ceases after suture tensioning and cushion filling.

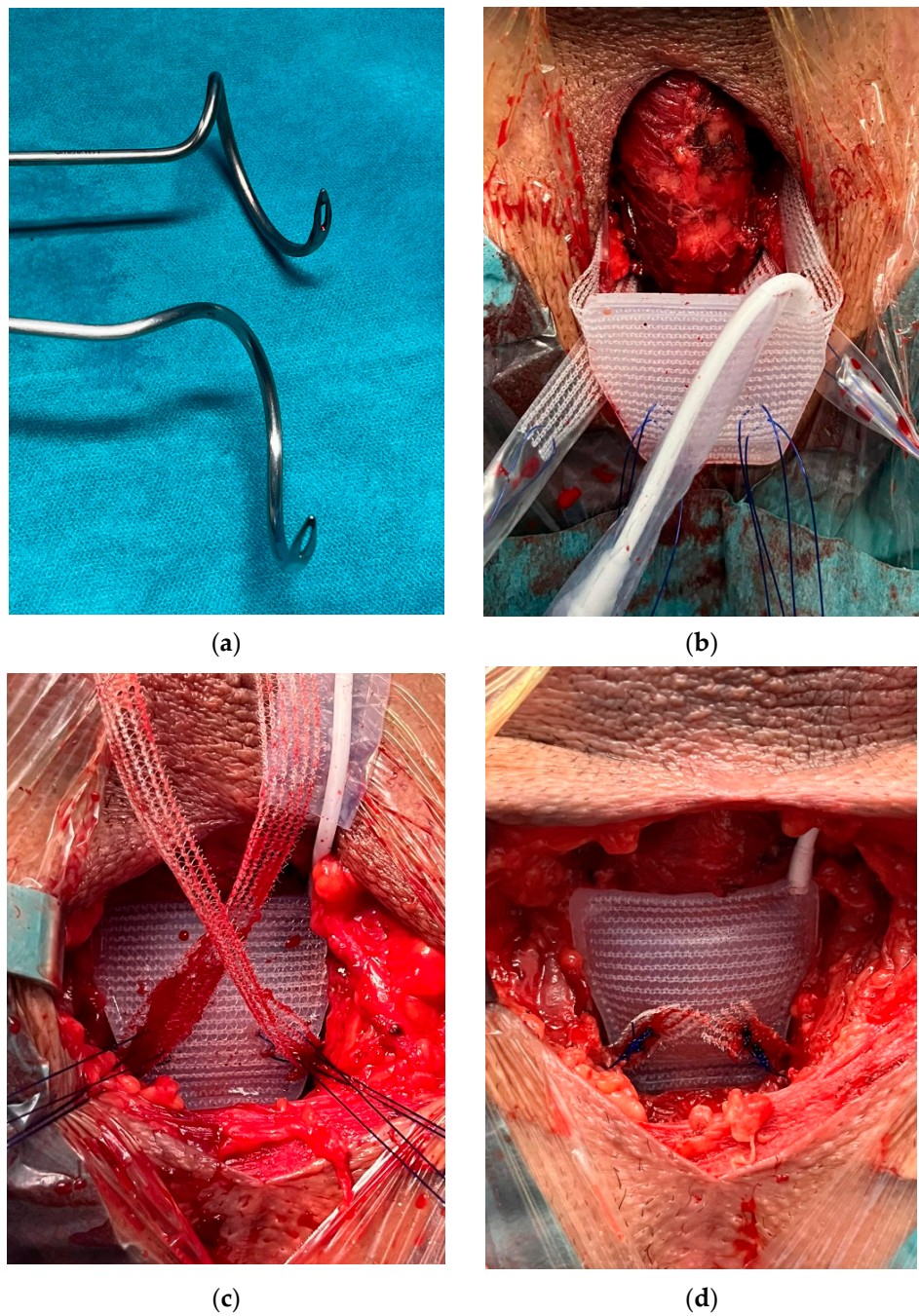

**Figure 2.** Main surgical steps of the adjustable trans-obturator male system (ATOMS) implant: (**a**) Passing needle for trans-obturator passage; (**b**) Implant brought into position by pulling the sling arms; (**c**) Sling arms are sutured to the device cushion, thus establishing a four-point fixation under the bulbar urethra; (**d**) Cushion is vented and filled to atmospheric pressure while the silicone covered port is left in the scrotum.

After removal of the plastic cover, the port is subcutaneously placed under the *Dartos* fascia, without tension, in a place accessible for trans-scrotal postoperative puncture if needed. Hemostasis must be undertaken because the perineal drain is not recommended. The surgical wound is closed, taking care not to puncture the port tube or the cushion. The patients are generally discharged several hours later or overnight with a Foley catheter in place until the first wound is cured in an office setting. Maintenance of the catheter for 2–3 days avoids wetting the incision and diminishes the risk of urinary retention. Analgesic medication is recommended.

If the patient is not continent postoperatively, filling of the scrotal port with saline solution is performed 1–2 months later. Subsequent additional filling with smaller volumes can be performed every 3–4 weeks until continence or maximum total filling (25–30 mL) of the system is reached. In case of retention, percutaneous removal of solution can be performed to facilitate voiding. Postoperatively, patients are evaluated every 3 months for a year and thereafter every 6 months, in consonance with their follow-up of prostate cancer.

## 5. ATOMS Results

A good number of studies on ATOMS and postprostatectomy incontinence have been published in peer-reviewed journals in recent years. We have identified 54 articles from database searching and other sources. Among them, operative results with ATOMS after radical prostatectomy have been identified in 48 articles.

Some are single-center, and others multicenter studies, and most of them are prospective evaluations. Not only the quality of the articles on the topic has increased, but also the level of evidence reported, as there are three systematic reviews and meta-analyses available on the subject [13,17,18] and one prospective comparative study performed between ATOMS and AUSs [10]. Additionally, two studies describe mean 5-year follow-up evaluation [29,31]. Some of the more recently published focus on different specific topics, such as urodynamics and a special population of patients. Among them, two studies evaluate the role of ATOMS after a failed AUS [14] and after failed retro-urethral sling [15]. Other studies specifically focus on complications and device durability. The largest study involves 902 patients from Europe and Canada and is dedicated to rescue surgery after failed ATOMS implant [32]. In summary, the body of evidence regarding the mode of action, surgical technique and operative and postoperative results, including not only efficacy and safety but also patient-reported outcomes (PRO) and satisfaction with the device, has been increasing in the past decade. However, no randomized study has been developed so far to compare ATOMS to other devices for PPI.

In spite of the existing large body of evidence revealed, there is some confusion caused by the results presented in older studies which often include the current SSP device together with other generations of devices (IP and SP). Studies in which pre-attached scrotal port predominates display better results and also reduce the rate of complications and device explantation [13,18]. We, therefore, decided to perform this non-systematic review with emphasis on continence, patient-reported outcomes, complications and device durability, focusing exclusively on the operative results of the SSP ATOMS device. Figure 3 shows the flowchart of studies included in this review. Eligibility criteria are based on the inclusion of articles with patients predominantly or exclusively treated with an SSP device, the only generation of the device that is currently available. The full-text articles excluded case reports, expert opinions or commentaries without specific data reported (*n* = 6), studies with patients who underwent intervention before 2014 (IP or SP ATOMS; *n* = 10) and studies with incontinence after transurethral resection of the prostate (TUR-P; *n* = 2). Forty-two studies with SSP ATOMS were included in qualitative synthesis, and that included systematic reviews (*n* = 3), articles partially overlapping with other previously published studies (e.g., follow-up or series updates; *n* = 9), and studies focusing on specific populations (*n* = 8). Only articles with no-overlapping data revealing outcomes of SSP ATOMS were included in the quantitative synthesis of results (*n* = 10).

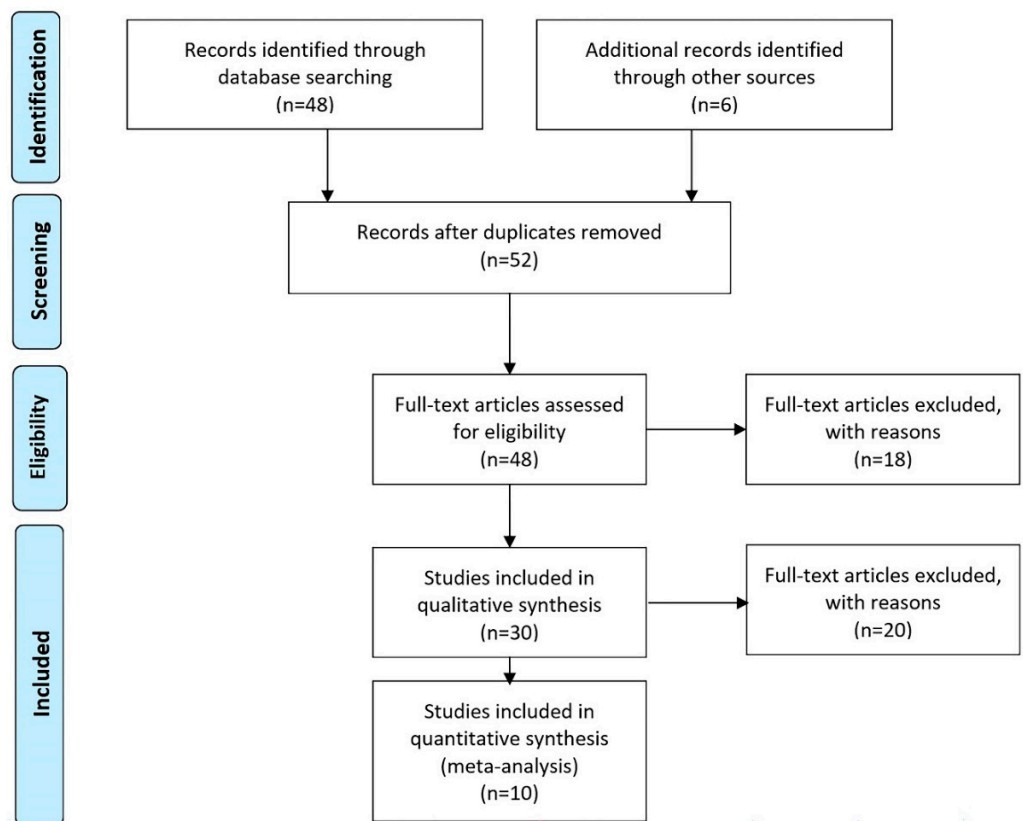

**Figure 3.** Flowchart defining studies included in this non-systematic review.

### 5.1. Efficacy Results with SSP ATOMS

Table 1 shows the pooled data of 1515 patients from the 10 studies with SSP ATOMS that focused on the evaluation of results [12,24–26,33–38].

**Table 1.** Studies with silicone-covered scrotal port ATOMS were included in the quantitative synthesis.

| Reference | *n* | Dry Rate (%) | Improved Rate (%) | Complications Rate (%) | Explant Rate (%) | Baseline PPD | Postop PPD | Prostatectomy (%) | Radiotherapy (%) |
|---|---|---|---|---|---|---|---|---|---|
| Buresova et al. 2017 [33] | 35 | 62.9 | 100 | 20 | 2.9 | 5 | 1 | 97.1 | 25.7 |
| Friedl et al. 2017 [25] | 287 | 64 | NA | 7 | 19.5 | 4 ± 1 | 1 ± 1 | 86.4 | 23.3 |
| Manso et al. 2018 [34] | 25 | 64 | 100 | NA | 0 | 4.8 ± 2.9 | 1.6 ± 2 | 92 | 44 |
| Angulo et al. 2018 [24] | 215 | 80.5 | 85.1 | 15.3 | 3.2 | 3.9 ± 2 | 0.9 ± 1.5 | 92.1 | 20 |
| Esquinas et al. 2018 [35] | 60 | 81.7 | 93.3 | 18.6 | 1.7 | 5 ± 3 | 0 ± 1 | 90 | 8.3 |
| Doiron et al. 2019 [26] | 160 | 80 | 87.8 | 22.3 | NA | 4 ± 1 | 0.5 ± 0.5 | 94.9 | 31.3 |
| Mühlstädt et al. 2020 [36] | 197 | 80.2 | NA | 27.3 | 6.4 | 6.1 ± 3.6 | 0.8 ± 1.3 | NA | 21.9 |
| Giammò et al. 2020 [37] | 98 | 79.6 | NA | 33.7 | 4.2 | 4 ± 2 | 1 ± 1 | 94 | 14.3 |
| Redmond et al. 2021 [38] | 289 | 73.2 | 89.3 | 23.5 | 7.9 | 4.2 ± 2.1 | 0.9 ± 1.5 | 100 | 33.9 |
| Dorado & Angulo 2022 [12] | 149 | 76.5 | NA | 22 | NA | 5 ± 3 | 0 ± 1 | 100 | 14.1 |
| Pool data | 1515 | 63–82 | 85–100 | 7–33 | 0–19 | 3.9–6.1 | 0–1.6 | 86–100 | 8–44 |

PPD: pads/day; NA: data not available.

The percentage of patients treated with radical prostatectomy in these studies ranged from 86–100%, as some included patients treated with radiotherapy with or without transurethral resection of the prostate. Also, the percentage of patients receiving radiotherapy was very variable and ranged from 8 to 44%.

Treatment with SSP ATOMS resulted in a 63–82% dryness rate. Dryness was defined as the use of no pad with one security pad allowed in eight studies and a <10 mL daily pad test in two studies. The improved rate was defined in six studies as >50% improvement in urine loss compared to baseline and ranged between 85 and 100%. Mean follow-up at the time the study was reported ranged from 9 to 45 months. The mean total number of system fillings ranged from 1 to 4.3 per patient. Regarding the magnitude of the pad count change reported, the median baseline pad count ranged from 3.9 to 6.1 pads per day (PPD), and the median postoperative pad count after adjustment ranged from 0 to 1.6 PPD.

Several studies have consistently reported that radiotherapy and baseline severity are factors associated with worse results [24–26,29,37]. In addition, radiotherapy has also been associated with a higher risk of device explantation [38]. Taking into account the baseline pad test, the Male Stress Incontinence Grading Scale (MSIGS) on standing cough test and radiotherapy, a nomogram to calculate the risk of ATOMS failure has been recently proposed [12]. However, there is no clinical study exclusively conducted on patients with PPI treated with radiation, and this will be really welcome in order to stratify risk factors of failure in these patients. Some professionals consider the option of ATOMS in this population more than other types of slings because radiotherapy is considered a contraindication for retrobulbar slings [39], and there is some controversy about whether it worsens the results of AUSs [40,41].

Other factors like diabetes, previous incontinence surgery and previous urethral surgery may also be related to worse efficacy results for ATOMS [25,26,37,38]. Very recently, a study focusing on patients with PPI and previously treated urethral stricture or bladder neck contracture has confirmed the effectiveness of ATOMS is reduced to 38% in this group of patients compared to 82.8% in patients without a history of stricture ($p < 0.0001$) [42]. However, using propensity score matching, the differences were maintained but did not reach statistical significance ($p = 0.236$). Additionally, perceived satisfaction is very similar to that of patients without a history of stricture [42]. Taking these facts into account, SSP ATOMS remains a very interesting alternative to trans-corporeal AUSs for PPI in patients with a fragile urethra, i.e., patients with prior sling failure [15], former removal of an AUS [14] or previous urethroplasty [42]. Also, the dual implant of ATOMS and a penile prosthesis appears feasible and effective [43].

Patient-reported outcome measurements (PROMs) must also be taken into account when analyzing the results of PPI treated with ATOMS. In a multicenter study, PROMS were evaluated in 181 patients after ATOMS adjustment. The proportion of patients that self-declared as being satisfied with the procedure was 87%, i.e., a proportion higher than that of patients achieving dryness [44]. More than 90% perceived their situation as better than before ATOMS implantation, and almost half of the patients ranked it as very much better than before. Among those achieving maximal satisfaction, several factors could be identified as independent predictors: total continence after device adjustment, baseline severity of incontinence, low postoperative pain at discharge and lack of complications [44].

*5.2. Safety Results with SSP ATOMS*

Table 1 also reveals the complication rate registered in 9 of 10 studies included in the quantitative synthesis, which ranged from 7 to 33% and were uniformly registered following Clavien Dindo classification within 90 days postoperatively [12,24–26,33,35–38]. The percent of explant rate ranged between 0–19%, depending mainly on the length of follow-up and in different patterns of rescue treatment proposed. A multicenter study with a large number of patients revealed an 8.3% explant rate after 4 years of follow-up, and the main reasons were persistent bothersome incontinence and scrotal erosion at the port site [32]. Another study exclusively focused on surgical complications and revealed they presented more frequently in the population of patients with previous radiotherapy ($p = 0.003$) and in patients with previous surgery for urethral stricture ($p = 0.017$) [36]. Intraoperative severe complications are very rare, and mechanic device failures are almost absent [13,24].

Postoperative pain is one of the most bothersome complications of ATOMS and can severely impact patient satisfaction. The median visual analog scale of pain (scale 0–10) in the first postoperative week was 2.4 ± 2.5 (range 0–7) in the Iberian multicenter study [24]. Perineal, scrotal or inguinal pain is a relatively frequent complication but very rarely a cause of surgical revision. It can be managed conservatively with oral analgesics in most cases [22,44]. Some degree of postoperative pain was registered in 15.4% of the patients undergoing surgical revision of ATOMS, but persistent pain was the main cause of revision in only 2.6% of the patients, and chronic pain is the leading cause of explantation in less than 1% of the patients [32].

The most severe complications are device infection and port erosion, often leading to device revision and explantation [32]. Although the ATOMS implant can be considered a relatively easy surgery, accurate technique and following preventive hygienic measures like double gloves and antibiotic wound washing seem advisable [11]. The reported incidence of ATOMS infection in multicenter studies is 2.7–6.2% [38,45], which is somewhere in the middle between the 1% reported for retro-urethral slings and the 8–10% admitted for AUSs (with or without urethral erosion) [46,47]. This could also be because ATOMS is partly a mesh with a silicone addition, and a single incision is needed for placement instead of the two incisions often used for AUS placement [10].

Device associated-infections typically occur months to years after the implant and are due to intraoperative bacterial colonization, sometimes facilitated by unusually excessive operative time. This contamination generates biofilm coating of the device, and infection is clinically evidenced after the transition to planktonic bacterial status. Often there is a long interval between implantation and the clinical manifestation of infection [45]. Another source of infection is early wound infection of perineal hematoma in cases with severe intraoperative bleeding. By contrast, device infection has never been reported as associated with repeated percutaneous postoperative port filling [45]. Positive urine culture at the time of the implant can be a risk factor for device infection [48]. Smoking habits or other individual characteristics remain unstudied.

Device infection tends to manifest as scrotal port erosion, the same as an AUS device infection manifests as urethral cuff erosion. In fact, scrotal erosion is the most frequent severe complication of ATOMS, and although it can be managed conservatively sometimes, it often leads to the explantation of the device [32].

### 5.3. Long-Term Durability of SSP ATOMS

Several studies have evaluated the device in the long term. A very interesting study showed the 5-year experience with ATOMS in a single institution in Germany [31]. The first multicenter European study reported 287 patients, but only 12.5% of them were with the SSP. The explant rate was 2.7% for SSP ATOMS and 21.9% for the rest, probably due to a very different dryness rate in each group. In fact, the dry rate was 81% for the SSP group compared to 51–68% for the rest [25]. Another early multicenter study, also with only 14% SSP ATOMS, confirmed a better postoperative ICIQ-SF score and lower complication rate for the SSP [22].

Conversely, the Iberian multicenter study mostly included SSP ATOMS generation. The 5-year mean analysis revealed 155 of 215 (72%) patients were dry at the last follow-up visit. Of these, 46% used no pads, and 26% used a security pad with a pad test result <10 mL. Taking into account the population achieving dryness, 94% remained dry at year 1, 91% at year 3 and 89% at year 5. The proportion of explanted devices was 11.6%, mainly due to inefficacy and scrotal port erosion [29]. Cox regression revealed that complications, baseline severity >5 PPD and irradiation were determinants of earlier explantation [29]. Very recently, the long-term survival rate of ATOMS has been demonstrated again by Giammò and Ammirati at a median follow-up exceeding 60 months [49].

A recently published nomogram to evaluate continence outcomes in patients with SSP ATOMS incorporates a standing cough test and baseline pad test. This study identified that patients with a daily pad test result <900 mL and an MSIGS of not 4 (i.e., early and

persistent stream or urine loss) are best candidates for SSP ATOMS [12]. Future analysis will likely confirm the best durability of this cohort of patients as well.

There is a very interesting publication dealing with therapeutic alternatives when ATOMS fails because of ineffectiveness or complications leading to device removal [32]. A second implant can be performed either simultaneously or at a deferred stage in cases with persistent bothering incontinence and always delayed in cases of infection or port erosion. The options are a second ATOMS implant or an AUS. Results are equivalent in terms of postoperative pain and complications but favor ATOMS in terms of operative time and continence results [32].

### 5.4. Comparative Studies Available with Other Devices to Treat PPI

Until now, no prospective controlled randomized trial has been developed between ATOMS and other incontinence devices. So, strictly speaking, there is no firm evidence to consider adjustable slings are superior to fixed male slings, although the indication for an adjustable or a fixed retrobulbar sling remains different. Fixed sling tends to be used in patients with limited urine loss, and radiotherapy is generally considered a contraindication for this incontinence surgery [39]. Adjustable slings were developed to cover a wider range of incontinence because intraoperative (i.e., Argus system) or postoperative (i.e., ATOMS, ProACT or the male REEMEX system) adjustment may correct mild-to-moderate PPI, and even severe incontinence in selected cases [50].

There is some evidence to compare the adjustable devices, despite prospective controlled studies being lacking. A systematic review and meta-analysis with 3059 patients reported ATOMS was superior to ProACT in terms of mean dryness rate (68% vs. 55%), overall improvement (91% vs. 80%) and satisfaction rate (87% vs. 56%), with a lower mean number of filling adjustments (2.4 vs. 3.5) and lower post-operative pad use per day (1.1 vs. 2.1) [17]. However, a very recent report from a single institution with long-term follow-up after ProACT implant, not included in that systematic review, reveals that careful patient selection and surgical experience may achieve high patient satisfaction in the long term [51]. Another systematic review and meta-analysis with 1919 patients revealed that ATOMS was superior to REEMEX regarding mean-dryness rate (69.3% vs. 53.4%) and improvement rate (90.8% vs. 80.2%), with a lower complication rate (18.9% vs. 35.8%) [18]. Postoperative adjustment with ATOMS is performed by simple percutaneous injection of serum into the scrotal port, but for REEMEX, an intervention is required to find and tighten the tension regulator of the device.

Only one prospective comparative non-randomized study has been developed to compare ATOMS and AUSs in a single institution for patients with moderate-to-severe PPI [10]. Differential pad test was lower for ATOMS than AUSs (−470 vs.—1000 mL), but total dryness (76.5 vs. 66.7%), social continence (90.2 vs. 85.2%) and satisfaction (92.2 vs. 88.9%) were similar, and so was the postoperative complication rate (22.6 vs. 29.6%). The surgical revision rate (6.9 vs. 22.2%) was lower for ATOMS because urethral erosion and urethral atrophy do not occur after ATOMS implant. Again, these figures must be taken into account with care because of the absence of randomization.

Regarding cost issues, AUSs are significantly more expensive than male slings. The cost differential in a 13-year period has been calculated as 12,643 (5863–26,557) dollars for AUSs and 10,429 (4877–16,439) for slings ($p = 0.0024$). This may explain, at least in part, the trend that slings were more commonly used than AUSs in the same total period (47.5% vs. 35.2%) [52].

## 6. Conclusions

ATOMS is an excellent alternative to treat PPI in patients with some degree of residual sphincter activity. Increased clinical experience and an accumulated body of evidence with this device have been notable during the past decade. Since the incorporation of silicone-covered and pre-attached scrotal ports in 2014, the device remains unchanged, and the surgical technique has been eased. Postoperative results with this device after adjustment

are very satisfactory: dry rate: 63–82%, improved rate: 85–100%, complication rate: 7–33%, device infection rate: 2.7–6.2% and explant rate: 0–19%. Also, the durability of the device is reassuring, with 89% that remain in place 5 years after implantation. An important limitation of this review is that the studies included varied significantly as some studies were prospective, and others were retrospective or collections of pooled data.

Despite the absence of randomized controlled studies, the literature findings confirm results of SSP ATOMS appear equivalent to those of AUSs in terms of continence, satisfaction and complications, but with a lower rate of revision in the long term. There are also studies suggesting that the ATOMS is an excellent alternative to treat PPI in special situations like failed fixed retrobulbar sling, previous urethral stricture and bladder neck contracture or dual implant with penile prosthesis. More consistent data to evaluate its role in the older population of patients at higher risk of impaired cognitive function and after radiation treatment for prostate cancer are needed.

**Author Contributions:** Conceptualization, C.T. and J.C.A.; methodology, C.T.; software, C.T.; validation, C.T., J.S., M.V.-C., I.A. and J.C.A.; formal analysis, J.C.A.; investigation, C.T., J.S., M.V.-C., I.A. and J.C.A.; resources, J.C.A.; data curation, J.C.A.; writing—original draft preparation, C.T.; writing—review and editing, C.T., J.S., M.V.-C., I.A. and J.C.A.; visualization, C.T.; supervision, J.C.A. All authors have read and agreed to the published version of the manuscript.

**Funding:** This research received no external funding.

**Institutional Review Board Statement:** The study did not require ethical approval. The study was conducted in accordance with the Declaration of Helsinki.

**Acknowledgments:** Alberto Alós Barco for photographic assistance.

**Conflicts of Interest:** The authors declare no conflict of interest.

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
