# Peer review of "Update on Adjustable Trans-Obturator Male System (ATOMS) for Male Incontinence after Prostate Cancer Surgery"

_curroncol, doi:10.3390/curroncol30040316_

Round 1
Reviewer 1 Report
There are 10 studies included in this study, with the attempt to formulate a systematic review of the topic.
Unfortunately, as the authors have indicated, there is too little substantial studies to formulate a conclusion. While this may be in part because the nature of correction fundamentally involves treatment of a complication to another treatment, it still should be even more strict in application of its inclusions.
Unfortunately, of the 10 studies there is significant overlap by the same authors, allowing repeated representation of the same study group. As such, the outcome is overrepresented.
The included studies also vary significantly in technique of investigation, disallowing collation of the pooled data. Some studies are prospective, another is retrospective, another is a collection of pooled data in itself.
Reviewer 2 Report
I would like to congratulate the authors on this fine review. Offers a comprehensive review of the available literature on this matter.
However, it might be good to show the results of comparison with other adjustable slings, since it is only mentioned at the end of section 1.
Since the AUS is the standard for treatment of PPI, it would be useful to compare the mentioned postoperative results to AUS results, as well.
Reviewer 3 Report
This review analyzes the current literature on this anti-incontinence device with intention to value the effectiveness, safety and duration of the silicone-coveredscrotal port (SSP) adjustable trans-obturator male systemATOMS, the only generation of the device that is currentlyavailable. Despite the absence of randomized controlledstudies, the literature findings confirm results of SSP ATOMS appear equivalent to those of AUS in terms of continence, satisfaction and complications, but with a lowerrate of revision in the long-term.
My suggestions:
- English language should be improved in both grammarand syntax.
- You should adda s keyword AUS (artificial urinarysphincter)
- Given that the outcomes between the ATOMS and AUS systems are similar in terms of safety and efficacy, I suggest analyzing any differences in the costs of the two devices
- compare the atoms system also to other devices such asProACT, in this regard i suggest the following articlehttps://pubmed.ncbi.nlm.nih.gov/35390245/
- Integrate the data obtained in this study on 1515 patients with other larger studies, at this regard i suggest this articlehttps://pubmed.ncbi.nlm.nih.gov/35729329/
Round 2
Reviewer 1 Report
acceptable changes
Reviewer 3 Report
Authors answered all comments and suggestions.